# Sleep Characteristics in Dogs; Effect on Caregiver-Reported Problem Behaviours

**DOI:** 10.3390/ani12141753

**Published:** 2022-07-08

**Authors:** Carrie Tooley, Sarah E. Heath

**Affiliations:** Behavioural Referrals Veterinary Practice, Chester CH2 1RE, UK; office@brvp.co.uk

**Keywords:** sleep, sleep duration, REM, behaviour, sleep disturbance, emotional health, arousal, dog, canine

## Abstract

**Simple Summary:**

Sleep duration and quality can be difficult to assess. Clinicians working in the field of veterinary behavioural medicine with dogs showing problem behaviours currently have little evidence-based literature to guide recommendations on assessment of sleep or treatment options if sleep is deemed poor. This study aims to broaden the level of knowledge regarding canine sleep durations and characteristics and begin research into the relationship between behavioural responses and the duration and quality of sleep. A questionnaire was used to capture information regarding canine sleep characteristics and caregiver perceptions of the severity of problem behaviours shown by this cohort. Responses regarding 1330 dogs were received and assessed. Dogs shown to sleep less than 6 h whilst their caregivers are in bed showed a greater caregiver-reported severity of problem behaviours. Dogs more easily disturbed from sleep at times their caregiver was out of bed, showed increased reported severity of problem behaviours. Whilst it is not possible to determine an optimal canine sleep duration, sufficient evidence is presented to argue that addressing problem behaviours by recommending increased activity may not be appropriate when it results in deprivation in relation to species-specific sleep requirements.

**Abstract:**

Optimal sleep duration and quality is difficult to define. There are strong arguments for a relationship between sleep, in particular REM sleep, and emotional health and behaviour in a variety of species. This study aims to broaden the level of knowledge regarding canine sleep durations and characteristics and begin research into the relationship between behavioural responses and the duration and quality of sleep. A caregiver questionnaire was used to capture information regarding the duration and characteristics of canine sleep, how easily this cohort of dogs were disturbed from sleep, and caregiver perceptions of the severity of problem behaviours shown by this cohort (*n* = 1330). A quadratic relationship between canine sleep duration whilst a caregiver is in bed and severity of problem behaviour is shown, with less than 8 h sleep and more than 10 h sleep correlating with increased severity of problem behaviours in this cohort. Dogs which were more easily disturbed from sleep at times their caregiver was out of bed, showed increased reported severity of problem behaviours. Whilst it is not possible to determine an optimal canine sleep duration, sufficient evidence is presented to argue that problem behaviour should not be remedied by sleep deprivation.

## 1. Introduction

Optimal sleep duration and quality is difficult to define. In humans, chronic inadequate sleep is a risk factor for physiological and emotional pathologies [1]. Sleep disturbance worsens quality of life and is associated with worsening perception of negative stimuli such as emotional challenges or chronic pain [2,3,4].

Sleep disturbance in both humans and canines has been shown to impair their visual recognition of emotional states demonstrated by human facial expressions [5,6] and increase the risk of confrontational behaviours in humans [7].

Parallels have been drawn between human and canine sleep both in physiological nature [8,9] and affective (emotional) and cognitive processing [10]. In the human field, there is evidence for an intimate and causal relationship between sleep and emotional brain function [11]. In the authors’ experience, and that of the caregivers of referred behavioural medicine patients, there exists a correlation between dogs diagnosed with anxiety disorders and those achieving a low duration of sleep and poor sleep quality. Characteristics defining poor sleep quality are vigilance during sleep and lack of cycling through Rapid Eye Movement (REM) and Non-Rapid Eye Movement (NREM) sleep. In human research, identification and assessment of REM sleep typically involves electrode placement and electroencephalography (EEG) readings [6] but this has rarely been replicated in dogs. In studies that have attempted EEG readings in dogs [12,13], habituation to the EEG equipment is not discussed. Sleep postures and locations under experimental conditions is not compared to those without EEG equipment. Observable REM sleep in mammals has been described as demonstrations of rapid eye movement, dreams (humans), oral movement, muscle twitching, and head twitching [14,15].

Whilst the physiological nature and potentially the purpose of sleep have strong similarities in humans and dogs, the pattern of the sleep–wake cycle varies significantly between the two species which can lead to challenges when the two species cohabitate. A high proportion of caregivers inadvertently interrupt the sleep of their canine companions [16]. In the experience of the authors, many caregivers also intentionally interrupt the sleep of their canine companion, out of misplaced concerned that canine sleep during human waking hours is due to boredom.

In addition to sleep duration, the nature of sleep achieved impacts emotional health and therefore behaviour in humans. There is a strong body of evidence in the human field to suggest that REM sleep plays an important role in emotional processing, supporting “affective brain homeostasis” in the brain to prepare an individual for emotional and social functioning in the day following sleep [11]. Reduced REM sleep in humans increases reported stress, anxiety, and anger in response to mildly aversive stimuli [17], increases impulsivity towards aversive stimuli [18], and increases the intensity of negative emotional arousal [19].

REM sleep in humans is thought to modulate emotional arousal via a noradrenaline reduction mechanism [20,21,22,23,24] meaning that an individual has a greater capacity to cope with emotional arousal of any sort (protective or engaging) the day after they achieve sufficient REM sleep during their night-time rest. This is of particular interest in veterinary behavioural medicine. In the authors’ experience, many patients show unwanted behaviours due to excessive emotional arousal, in part due to the engaging emotional systems (this is possible in dogs that bark excessively during play or are over-exuberant during well-intentioned social interactions).

There is a strong argument that REM sleep enables emotional memory processing [25,26,27,28]. This processing is a necessary requirement for retaining emotional memories which are important for survival, but without retaining the intense emotional experience along with that memory. For example, if an individual has an experience involving a near-miss with a lorry, for the rest of their life it is advantageous for them to recognise moving lorries as a potential threat. It is not advantageous to them to experience a full fear response motivating extreme avoidance behaviour simply when they are in proximity to, but not in the path of, a lorry.

Bunford et al. [12] discuss the negative impact of sleeping in locations other than a dog’s home on REM sleep. Lima et al. [29] describe various mammals using a relatively vigilant state of sleep when they perceive an increase in potential risk in the environment. Kortekaas and Kotrschal [30] show that dogs resting in isolation show more alertness to their surroundings than group-living dogs. These studies show the impact that the sensation of safety has on sleep duration and quality. Providing dogs are not woken from sleep, emotionally healthy individuals resting in environments they perceive to be safe, should achieve good quality and duration of sleep. Reduced quality and duration of sleep may occur in two contexts. Firstly, emotionally healthy individuals who assess an environment to be less safe will show higher levels of vigilance to potential threats (as humans might following a break in at the family home or if the family children are likely to trip over the dog in its normal resting place when they are home at the weekend). Secondly, individuals with persistent protective emotional states associated with generalised anxiety (intrinsic or adaptive) may misinterpret a safe environment as one requiring vigilance. In either cohort, this poor sleep may exacerbate the protective emotional state of the individual leading to a cycling of worsening sleep and more extreme protective emotional bias.

It is reasonable to suggest that a reduced quantity of canine sleep has emotional, cognitive, and behavioural repercussions similar to human sleep deprivation; however, no clear data exists on common canine sleep durations or the relationship between this and emotional state and behaviour. If caregivers interrupt canine sleep, this may have exaggerated consequences on mood, arousal, and behaviour in dogs. The effects could be disproportionate to the reduction in duration of sleep if the interruption results in a higher proportion of REM sleep reduction than Non-REM (NREM) sleep if, as in humans, the role of REM sleep in the dog is in part to support affective (emotional) homeostasis. It is important to note that the reason for low sleep duration or a lack of cycling through REM and NREM sleep is not determined in many of the studies described. Potential causes include protective emotional arousal (motivating vigilance to potential threats or disturbance of sleep due to pain) or inappropriate duration of engaging emotional arousal (such as engagement with a fun activity or vigilance towards caregivers to ensure a dog does not miss an opportunity to engage with play or receive a treat). Due to study methodologies, it is generally not possible to differentiate between engaging versus protective emotional states as the cause for reduced sleep. At population level, the cause for reduced sleep duration or quality is immaterial as it is the sleep duration which has the main effect on emotional and physiological health regardless of the cause. However, at an individual level, diagnosing the cause is an essential pre-requisite to the creation of an appropriate treatment plan.

In the absence of reliable data regarding appropriate canine sleep durations, it is a challenge for clinicians to determine if a dog is sleep deprived or to give recommendations to caregivers as to how much sleep their pet should be getting. 

This study assesses caregiver-reported duration of sleep and in-sleep behaviours displayed by pet dogs in the UK. Data regarding severity of behavioural problems (as reported by caregivers) is analysed and the relationship between sleep duration and reported severity of behavioural problems described. The impact of REM sleep on severity of caregiver-reported behavioural problems is also assessed.

## 2. Materials and Methods

### 2.1. Experimental Design

An online questionnaire (see Appendix A) was presented via Survey Monkey^®^. Initial dissemination of the questionnaire was performed via a single post with the questionnaire link in the first 20 Facebook^®^ groups which were listed following a search for “dog UK”. Groups supporting unethical practice (for example, ear cropping) were excluded. If group admin permissions were required prior to posting, permissions were sought. In addition, a one-time link via the researcher’s professional Facebook profile and a one-time link via the Facebook group of the business supplying the prize draw incentive (a mail-delivery toy package for dogs), was published. Snowball sampling was enabled as any Facebook user was able to “share” the survey link to their profile timeline, to other groups or to specific “friends” if they chose.

Data was collected via the Survey Monkey^®^ 2022 programme,)Momentive, San Mateo, CA, USA), handled via Microsoft Excel^®^ (version 2205, Microsoft, Redmond, WA, USA) and statistical analysis was performed using Minitab^®^ (version 19, Coventry, UK.)

The caregiver questionnaire (Appendix A) gathered information regarding sleep achieved by dogs whilst their caregivers were in bed (question 9) and out of bed (question 10). Each response was the caregivers’ best estimate of this duration. This data was recorded in seven two-hour-interval categories for each context (up to 2 h, 2 h–3 h 59 min, 4 h–5 h 59 min, 6 h–7 h 59 min, 8 h–9 h 59 min, 10 h–11 h 59 min, 12 h or more).

Dogs which have been thought to enter REM sleep on at least one caregiver-witnessed occasion were allocated to the REM sleep cohort. They were selected from respondents answering question 11 (see Appendix A) “What signs indicate to you that your dog is asleep?” that ticked “They twitch or vocalise as though dreaming” alone, or in addition to other options. The lacking-REM sleep cohort were selected as respondents answering the same question that ticked any of the other responses but did not tick “they twitch or vocalise as though dreaming”. 

Caregiver’s perceived severity of each dog’s problem behaviour was recorded as the “problem behaviour score” (PBS) from their response to Question 16 “On a scale of 0 (this dog is perfect and shows no problem behaviours) to 10 (this dog behaves in a way I cannot tolerate), please give your opinion on the severity of the behaviour this dog shows” (see Appendix A). The online question format offered a slider scale and Survey Monkey^®^ software converted this to a 0–100 scale.

### 2.2. Statistical Analysis

The statistical analysis was processed by Minitab^®^. A general linear model (GLM) with PBS as the response variable was performed with the Minitab^®^. Statistical significance was set at *p* < 0.05.

Factor analysis was performed using Minitab^®^ (version 19, Coventry, UK) to investigate whether questions 13 and 14 of the caregiver questionnaire (Appendix A) could be combined. These questions gathered information about a dog’s typical response to within home stimulation (a caregiver entering or leaving a room) and external stimulation (a noise out of the home which the caregiver can easily hear) which occurs whilst they sleep. Responses were recorded on a Likert scale for a total of seven response options across the two stimulus examples.

## 3. Results

### 3.1. Demographic

Responses regarding 1330 dogs were received, with a distribution of entire females (*n* = 123, 9%), neutered females (*n* = 519, 39%), entire males (*n* = 180, 14%), and neutered males (*n* = 499, 38%) and 9 respondents (1%) not answering this question. In total, 143 breeds were represented with crossbreeds (*n* = 399, 30%) being the largest breed category. Mean dog age = 69 months, range = 6–211 months. 

A clear bias of caregiver (respondent) gender was present (female *n* = 1361, 96.6%, male *n* = 47, 3.3% and other/prefer not to say *n* = 2, 0.1%). A wide range of caregiver age ranges was represented (less than 19 *n* = 14, 1%, 20–29 *n* = 214, 15%, 30–39 *n* = 302, 21%, 40–49 *n* = 318, 22%, 50–59 *n* = 327, 23%, 60–69 *n* = 179, 13%, 70–79 *n* = 54, 4%, 80 or over *n* = 1, 1%).

### 3.2. Sleep Duration

Sleep durations in the two contexts (whilst a caregiver was in bed and whilst the caregiver was out of bed) are described in Table 1 and Figure 1 and Figure 2. Sleep duration achieved by dogs whilst their caregivers are in bed shows a smaller range, with no respondents reporting less than two hours sleep in this context and a right skew (mean category = 6 h–7 h 59 min sleep, *n* = 1329). In comparison, sleep achieved whilst caregivers are out of bed shows a full range of responses and a lower mean category (4 h–5 h 59 min sleep, *n* = 1329).

### 3.3. Sleep Characteristics

Respondents gave information regarding whether their dog had a choice to sleep with or without human company in question 12 of the caregiver questionnaire (Appendix A). The choice made by dogs (*n* = 1067) which “always” or “usually” had the choice whether to sleep with or without human company was analysed (Table 2). Of these dogs, most (*n* = 670) chose a room with human company the majority of the time.

The most commonly caregiver-observed sleep characteristic (Table 3) was a dog having their eyes shut, with twitching or vocalising as though dreaming (signs associated with REM sleep) being the second most commonly observed sleep characteristic. 

### 3.4. General Linear Model

No significant relationship was found between PBS and the following: age, sex, neuter status, breed, or whether the dog showed behaviours consistent with REM sleep (Table 4).

### 3.5. Influence of Sleep Duration Achieved by a Dog Whilst Caregiver Is in Bed on PBS

A statistically significant relationship was found between the sleep duration a dog achieves (as reported by the caregiver) whilst their caregiver is in bed and PBS (Figure 3). Quadratic terms were entered for the sleep duration predictors, to capture the possibility that there might be an optimal sleep quantity, rather than a monotonic relationship. Caregivers reported that below 8 ≥ 10 h, the less sleep a dog achieved whilst their caregiver was in bed, the higher the PBS. At the other end of the scale, the respondents in this study showed an increase in PBS with an increase in sleep duration over 10 ≥ 12 h, albeit with divergent confidence intervals.

### 3.6. Influence of Sleep Duration Achieved by a Dog Whilst Caregiver Is Out of Bed on PBS

An initial relationship between sleep duration achieved whilst a caregiver is out of bed and PBS appeared to show that the greater the duration of sleep a dog achieved in this context, the lower the PBS (Coeff = −1.187, *p* = 0.039). However, an impact of this specific sleep duration category on PBS was not demonstrated when an ease-of-disturbance factor was included in the model.

### 3.7. Factor Analysis

The factor analysis for questions 13 and 14 (regarding typical responses to caregiver and external stimuli which occur whilst the dog is asleep) showed two factors had more variance than would be expected from randomness, according to Horn’s parallel analysis. The first was named “ease-of-disturbance”. This combines the responses “wake and follow” for both caregiver and external disturbance, “wake and react” for external disturbance, and “stay asleep” (negatively weighted) for both caregiver and external disturbance. The second was named “wake-and-assess”, combining all questionnaire “wake and assess…” responses for both caregiver and external disturbance.

Following factor analysis, a small but statistically significant relationship between ease-of-disturbance and PBS was found (Coef = 3.87, *p* = 0.003) showing an increase in ease-of-disturbance correlated with an increase in PBS. There was no significant relationship between wake-and-assess and PBS. No relationship was found between ease-of-disturbance and whether REM sleep was observed.

## 4. Discussion

This study aimed to support current literature on sleep durations and characteristics of dogs by reporting on a varied cohort of dogs living in a home environment. Recent studies which report the duration of typical canine sleep [31,32,33] tend to assess “daytime” and “night-time” sleep. The interest in this study lay predominantly in caregiver availability and interactions, so the cohorts “whilst caregiver is out of bed” (assumed to be broadly equivalent to daytime) and “whilst caregiver is in bed” (assumed broadly equivalent to night-time) were used. The findings for sleep whilst caregiver is in bed from this study agreed with previous findings for night-time sleep for 12-month-old dogs living in a home environment [32] and shelter dogs of a broader age range [31,33]. Sleep durations whilst caregivers were out of bed varied from previously reported durations, with this study finding that most dogs slept 4 h–5 h 59 min in this context, in contrast to 3.0 h daytime sleep in 12-month-old dogs [32] and 14–37 min in the shelter context [33]. One limitation of this study is that caregiver assessment of sleep was recorded, including assessment whilst caregivers themselves were asleep. This caregiver-questionnaire generated data could lead to a higher rate of inaccurate responses when compared to either direct interviewing or independent observation (via video, for example). Offering sleep duration categories limited the specificity of the data collected and prevented total sleep duration over a 24 h period being assessed. 

When comparing this study cohort to the Kinsman et al. [32] cohort of 12-month-old dogs also observed when asleep by their caregivers in the home environment, similarities in sleep choices and sleep characteristics were evident. Dogs in this study who had a choice, chose to always (25.8%) or usually (37.0%) sleep in the same room as people. Similarly, 59% of dogs given the same choice, chose to sleep “close to people” in the Kinsman et al. study. The caregiver-observed sleep characteristics associated with REM sleep were reported at a similar rate in both this study (82.3%) and by Kinsman et al. (73.0% minimum).

No relationship was found between PBS and dog age, sex, neuter status, or breed. Current views in behavioural medicine literature vary on the potential influence of these factors on behavioural traits, in part due to the use of different scoring systems [34,35,36,37,38]. PBS is not a validated assessment tool used for research, thereby limiting direct comparison with previous work. It is based on a single item questionnaire answer. Free text responses regarding the nature of the problem behaviours eliciting the PBS were encouraged in the caregiver questionnaire and in the authors’ opinion the majority of these responses correlated with the reported PBS. Qualitative analysis of these free text answers was not possible due to an insufficient level of detail. Future research into this area would benefit from a more detailed assessment of presenting problem behaviours; to include clinician assessment in addition to caregiver perception of the presenting problems. PBS remains a valid choice in this initial exploratory study as a clinically relevant factor to both the individual patient and their caregivers.

No significant relationship was found between PBS and caregiver observation of REM sleep. The methodology of this study is likely to have influenced this, with a very low level of detail captured. Use of studies and EEG to accurately assess REM sleep in dogs brings the potential limitation that placement of EEG electrodes may influence canine sleep behaviour. However, within a caregiver questionnaire format, more specific information could be requested including REM sleep behaviour duration and pattern of sleep behaviours associated with REM sleep. There is a strong theoretical argument for the influence of REM sleep over emotional health and behaviour, but studies designed to specifically answer this question may need to gather much more detailed information from a smaller cohort to begin this work. Direct observation via video or audio recording of dogs sleeping in their home environment may be optimal.

PBS was shown to increase with dogs sleeping less than 8 ≥ 10 h whilst their caregivers are in bed. The less they sleep below 8 ≥ 10 h, the higher the PBS. It was beyond the scope of this study to thoroughly assess the nature of the problem behaviours each respondent reported. It is therefore possible that night-time activity which disturbs caregiver sleep could in some cases have contributed to an increased PBS. However, in the experience of the authors, dogs which show poor sleep quality overnight tend to also show a variety of other problem behaviours associated with poor control of emotional arousal. The correlation between poor sleep duration whilst a caregiver is in bed and PBS does not explain causation. It is possible that emotionally healthy individuals may show poor sleep duration due to justified night-time disturbance, for example, when living with a baby that cries through the night. The PBS of these individuals may fluctuate; increasing during phases when sleep is disturbed and decreasing when sleep duration is adequate. In these cases, the low sleep duration is the main cause of the raised PBS. Equally, it is possible that individuals with a strong protective emotional bias (for example, due to generalised anxiety) may rest in a more vigilant state and be more likely to be disturbed by stimuli which would not disturb an emotionally healthy individual. In this context, protective emotional bias may be the main common cause for both poor sleep duration and high PBS. When considering sleep durations greater than 10 h whilst a caregiver is in bed, the relationship with PBS is unclear. In this cohort, the PBS increases with longer sleep duration. However, the lower confidence interval shows a decreasing trend whilst the upper follows an increasing trend. This means we cannot predict from this data set, whether sleep durations above 10 h lead to an increased or decreased PBS in the general population. Potential reasons for sleep durations greater than 10 h correlating with an increased PBS were considered. A causal relationship whereby longer periods of sleep lead to lack of appropriate mental stimulation could lead to problem behaviours. Alternatively, an individual may sleep for extended periods due to ill health or pain, both of which can also be causal factors in problem behaviours.

An initial trend seemed to show that the greater the duration of sleep a dog achieved whilst their caregiver was out of bed, the lower the PBS. This supports the clinical experience of the authors. A parsimonious explanation would be that the more time a dog is asleep, the less waking hours they have in which to display unwanted behaviours. Alternatively, the likely benefit of sleep for emotional health, means that it is plausible that dogs able to sleep well the majority of the time, show a more engaging emotional bias. Good quality and duration of sleep enhances an individual’s ability to lower their emotional arousal and deal with future challenges appropriately [11]. Ultimately, when ease-of-disturbance was included in the model, the influence of sleep duration whilst a caregiver was out of bed on PBS was no longer demonstrated. Whilst the data indicates ease-of-disturbance to be the main influencing factor, the underlying influences on both this and sleep duration may relate to the emotional health of an individual and, in turn, their ability to reduce emotional arousal during rest.

Factor analysis associated with stimuli which disturb a dog’s sleep, and the associated responses of those dogs showed clinically interesting results. Two clear factors emerged which were named ease-of-disturbance and wake-and-assess, though only the former showed a significant effect on PBS. Dogs with a high ease-of-disturbance score would react to both social stimuli (caregiver entering or leaving the room the dog is sleeping in) and environmental stimuli (a noise external to the house which is easily heard by the caregiver) by either finding or following their caregiver or reacting towards the environmental noise. Dogs with a low score within this factor would remain asleep during exposure to both social and environmental stimuli. A greater ease-of-disturbance score correlated with increased PBS which supports the authors’ clinical experience. It is likely that individuals who are more easily disturbed experience a more vigilant state of sleep than those who are difficult to disturb. The cause for that vigilant state of sleep may be a more protective emotional bias, which can also motivate many of the behaviours that caregivers find difficult to tolerate.

The relationship between PBS and sleep duration whilst the caregiver is in bed should be interpreted with caution. Whilst the large cohort size leads to a statistically significant result, the degree of increase in PBS with each increase in sleep duration category is small and may not be clinically significant when considering the quality of life of the dog or their caregiver. Nevertheless, this data does show an interesting trend. Together with literature in the human field and strong anecdotal evidence in veterinary behavioural medicine, this trend indicates value in further research into the specifics of the relationship between sleep duration and problem behaviours.

It is important to consider whether caregiver perception (generating PBS) should be the measure by which we assess the potential link between sleep and emotional health. It is a useful broad-measure tool in this initial questionnaire study; however, emotional health is unlikely to fully correlate with caregiver-reported severity of problem behaviours. For example, dogs who in their caregiver’s opinion “behave perfectly” during a frightening experience will be in a negative emotional state but may select a behavioural response acceptable to the caregiver, such as inhibition or appeasement [39]. If this is a frequent occurrence, those dogs are likely to show a low PBS despite poor emotional health and compromised welfare. Equally, caregiver misunderstanding of normal canine behaviour could result in the reporting of context-appropriate species-specific behaviours, such as barking or digging, as being problematic. A behavioural medicine assessment would be needed to research the intricacies of the relationship between sleep duration and emotional health but may not be possible on a large scale. Research into sleep duration and behaviour within a cohort of behavioural medicine referral patients could prove an interesting starting point.

## 5. Conclusions

Data on sleep duration and characteristics within the home environment in a broad cohort of dogs was obtained. A relationship between sleep duration whilst caregivers are in bed and caregiver-reported severity of problem behaviours was shown. In addition, a relationship between sleep disturbance whilst a caregiver is out of bed and caregiver-reported severity of problem behaviours is described. Further research is required into sleep characteristics and duration, and the influence of both on emotional health and problem behaviours. There is sufficient support to argue that addressing problem behaviours by recommending increased activity may not be appropriate, when it results in deprivation in relation to species-specific sleep requirements. Current knowledge regarding canine sleep is insufficient to determine the optimal duration of sleep for emotional health in this species, with further research required in this area.

## Figures and Tables

**Figure 1 animals-12-01753-f001:**
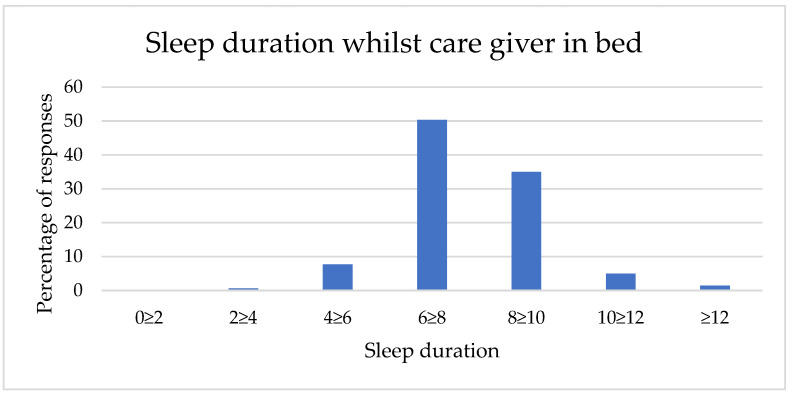
Caregiver-reported sleep duration of a dog whilst the caregiver was in bed.

**Figure 2 animals-12-01753-f002:**
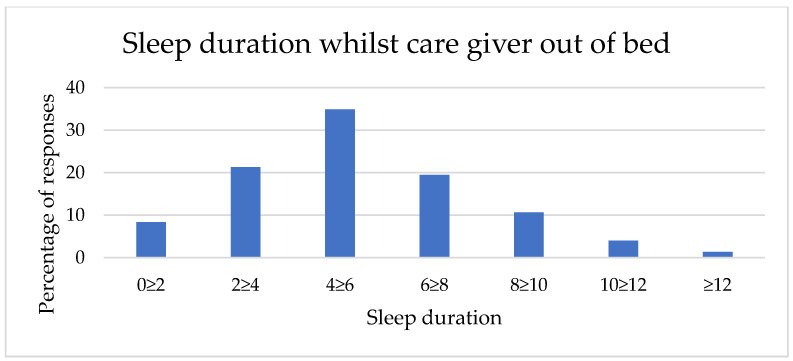
Caregiver-reported sleep duration of their dog whilst the caregiver was out of bed.

**Figure 3 animals-12-01753-f003:**
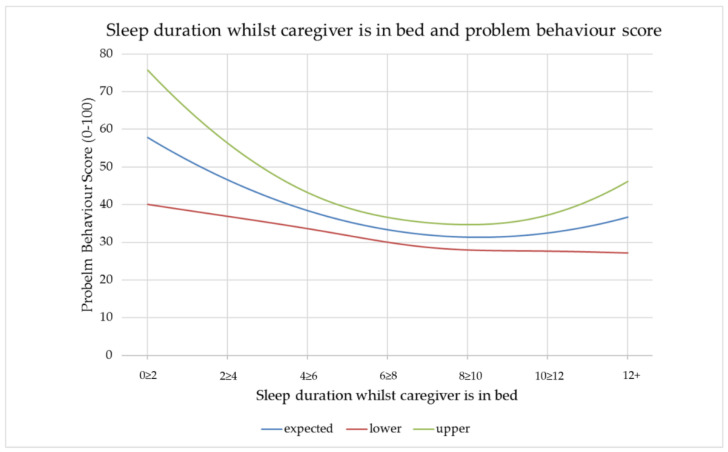
The relationship between sleep achieved by a dog whilst their caregiver is in bed and PBS. “Expected” represents the expected value from this sample. “Lower” and “upper” represent the bounds of a 95% confidence interval on these expected values.

**Table 1 animals-12-01753-t001:** Caregiver-reported duration of sleep that their dog achieves whilst the caregiver is either in bed or out of bed.

Sleep Duration	Whilst Caregiver Is in Bed		Whilst Caregiver Is Out of Bed	
	*n*	%	*n*	%
Less than 2 h	0	0	111	8
2 h–3 h 59 min	8	1	283	21
4 h–5 h 59 min	102	8	464	35
6 h–7 h 59 min	670	50	259	19
8 h–9 h 59 min	465	35	142	11
10 h–11 h 59 min	66	5	53	4
12 h+	19	1	18	1

**Table 2 animals-12-01753-t002:** Choices made by dogs regarding sleeping with or without human company.

This Dog Chooses to Sleep in a Room with a Human Family Member Present	Responses *n* (%)
Always	275 (25.8)
Usually	395 (37.0)
Sometimes	315 (29.5)
Rarely	44 (4.1)
Never	15 (1.4)
Unsure/Not applicable	23 (2.2)

**Table 3 animals-12-01753-t003:** A summary of caregiver observed sleeping behaviours (*n* = 1327 respondents).

Caregiver Observation	Responses *n* (%)
Their eyes are shut	1215 (86.0)
They twitch or vocalise as though dreaming	1163 (82.3)
They breath more slowly	1034 (73.2)
They are staying still	937 (66.3)
They go floppy/have relaxed muscle tone	856 (60.6)
They snore	807 (57.1)
They are unresponsive to stimuli that would normally cause excitement	552 (39.1)
Other	84 (5.9)

**Table 4 animals-12-01753-t004:** General linear model output for PBS.

Variable	Coeff	SE	F	df	*p*	VIF
Age (months)	−0.0288	0.0179	2.59	1	0.108	1.17
Sex	2.31	1.45	2.54	1	0.111	1.02
Neuter status	2.49	1.88	1.76	1	0.185	1.20
Breed			1.89	7	0.068	
Out bed	−1.187 *	0.576	4.25		0.039	1.09
Out bed squared	0.350	0.317	1.22	1	0.270	1.02
In bed	−1.780	0.937	3.61		0.058	1.03
In bed squared	1.549 *	0.703	4.85	1	0.028	1.02
REM	−2.45	1.90	1.67	1	0.197	1.02

Coeff = coefficient; SE = standard error; F = F value; df = degrees of freedom; *p* = *p* value; VIF = variance inflation factor; * = *p* < 0.05.

## Data Availability

The data presented in this study are available on request from the corresponding author.

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
