# Peer review of "Sleep Characteristics in Dogs; Effect on Caregiver-Reported Problem Behaviours"

_animals, 2022, doi:10.3390/ani12141753_

Round 1

Reviewer 1 Report

This was an interesting and well-written paper on a topic that could be of significant interest to people around the globe. Though I currently do not own a dog, I have lived with dogs in the past and have never considered the importance of sleep disruption on behavior and mood regulation. Clearly, this is an important consideration and the authors do well to set up the rationale of this study with the existing literature. 

My main concern, which is a significant one, is that the problem behavior score (PBS) is based entirely on the response to a single question (Q16 on a 0-10 scale). Single item scales are problematic, and given the importance placed of this dependent measure within the results, to me this presents a major limitation to the current study that should be acknowledged. 

Minor comment: REM is used as an abbreviation in the introduction before rapid eye movement is mentioned. 

Author Response

Thank you.

I am happy to review the discussion with this limitation in mind and present the conclusions in the context of this being an initial exploration of the relationship, with much more work to be done in the future. I suggest a change to Ll. 295-298 “PBS is not a validated…behavioural medicine consultations” to: “PBS is not a validated assessment tool used for research, thereby limiting direct comparison with previous work. It is based on a single item questionnaire answer. Free text responses regarding the nature of the problem behaviours eliciting the PBS were encouraged in the caregiver questionnaire and in the authors’ opinion the majority of these responses correlated with the reported PBS. Qualitative analysis of these free text answers was not possible due to insufficient level of detail. Future research into this area would benefit from a more detailed assessment of presenting problem behaviours. PBS remains a valid choice in this initial exploratory study as a clinically relevant factor to both the individual patient and their caregivers.”

Re: "Minor comment: REM is used as an abbreviation in the introduction before rapid eye movement is mentioned." - Thank you. Suggested amendment: Ll. 52 “…cycling through REM and NREM sleep” to read “…cycling through Rapid Eye Movement (REM) and Non-Rapid Eye Movement (NREM) sleep” with Ll. 69 “…suggest that Rapid Eye Movement (REM) sleep…” to read “suggest that REM sleep”.

Reviewer 2 Report

I greatly enjoyed reviewing the manuscript “Sleep characteristics in dogs; effect on caregiver reported problem behaviors”. The study is relevant to the field of animal behavior, in particular clinical animal behavior. The design of the research project has important limitations, but the Author discusses these limitations extensively and correctly.

I have mostly minor recommendations to improve the quality of the manuscript, which I will list at the end of the report, except for an apparent discrepancy between the results reported in the Abstract (ll. 28-30) and the results reported in the section 3.5 Influence of sleep duration achieved by a dog whilst caregiver is in bed on PBS (ll. 238-242). In the Abstract, the Author reports clearly that “less than 6 hours sleep and more than 10 hours sleep” correlates with increased PBS. In section 3.5, the Author reports that below 8 to 10 hours and above 6 to 8 hours sleep the PBS is higher.  This is at least my understanding. This is in apparent contradiction with what reported in the Abstract and displayed in Figure 3. Moreover, in the Discussion the Authors again reports that below 8 to 10 hours of sleep the PBS increases. Can the Author please clarify this apparent contradiction and present the results in a consistent manner throughout the manuscript?

Please find below some minor comments and recommendations:

-          Ll.49-52: the Author affirms that “anecdotal support exists…”. Can this support be documented with any reference at the end of the sentence? Maybe a book, conference proceedings, or an editorial?

-          Ll.52-57: Consider removing sentence “In human research… without EEG equipment”. I do not think this consideration is relevant to your study.

-          Ll.86-90: Consider removing sentence “For example…. of a lorry”. I do not believe it is necessary.

-          Ll.260-261: Need revision. “Combing” should be “combining”. I also think the sentence is incomplete.

-          Ll.320-321: Please review “These individuals… low sleep duration”. I think this concept could be expressed more clearly.

Author Response

Thank you for your kind comments.

Thank you for pointing out this discrepancy in reporting the results – I shall address this section-by-section.

ABSTRACT: This has occurred as the results section give the exact sleep duration categories and in simplifying this to report hours (rather than categories) in the abstract, the implication is incorrect. Ll 29 in the abstract can be changed to “less than 8 hours sleep and more than 10 hours sleep” to be consistent with the results reported in section 3.5.

SECTION 3.5: With the abstract change mentioned above, Ll 239 “…below 8≥10 hours…” is now consistent and correct. Ll 241 “…increase in sleep duration over 6≥8 hours…” is an error, thank you for noticing this. It should read “…over 10 ≥12 hours…”.

DISCUSSION: With the abstract change mentioned above, Ll 310-311 are now consistent and correct

Regarding each of your further points, please see my responses with reference to the lines you mention:

-          Ll.49-52: I am not aware of any referenceable material on this subject, but would be happy to change “anecdotal support” to “in the authors’ experience and that of the caregivers of referred dogs”

-          Ll.52-57: In my view, these three sentences are necessary for the following reasons; 1) as a comparison of sleep study methods in humans and dogs, 2) to offer a short critique on studies using EEG in dogs and 3) as a justification for why EEG was not attempted in this study. My preference would be leave this section unchanged, but if the reviewers feel strongly I would remove all three sentences.

-          Ll.86-90: The authors discussed this at length prior to first submission. We agree that with a behaviourally aware audience, the paragraph regarding the lorry is unnecessary. However, we hope this animals paper will reach a wider audience and in our experience, the use of analogies such as this are an important tool in enabling interested individuals with no prior behavioural medicine education to understand the concepts we discuss.

-          Ll.260-261: You are correct, thank you for picking this up. “combing” should read “combining all”

-          Ll.320-321: Suggested change: “The PBS of these individuals may fluctuate; increasing during phases when sleep is disturbed and decreasing when sleep duration is adequate.”

Reviewer 3 Report

The present study investigates the potential relationship between sleep duration and the severity of behavioral problems in dogs. The study highlights some important limitations about specific factors that need to be investigated further for a better understanding of this relationship.

I would suggest highlighting in the discussion the need to collect further data by direct observation of dog behaviour for both the sleep and the reported behavioral problem, which is only collected by the mean of questionnaires (and may not reflect the presence of a real behavioral problem but rather a behaviour considered as problematic by owners: see d’Ingeo, S., Ferlisi, G., Minunno, M., Palmisano, G. L., Ventriglia, G., Siniscalchi, M., & Quaranta, A. (2022). Motivations of Human Helping Behavior towards Dogs. Veterinary sciences, 9(3), 145.

and

d’Angelo, D., Ciani, F., Zaccherini, A., Tafuri, S., Avallone, L., d’Ingeo, S., & Quaranta, A. (2020). Human-animal relationship dysfunction: A case study of animal hoarding in Italy. Animals, 10(9), 1501.).

Moreover, an investigation regarding the type of population the study is referred to is needed. Therefore, I would suggest reporting (and analyse) the type of behavioral problems described by owners. Subjects may need to be divided in different groups according to the presence/absence of the problem (in order to understand if the “healthy” subjects may constitute a reliable control) and the relationship between the type of problem (e.g. anxiety, aggression) and the sleep duration should be investigated.

Author Response

Re: your point on questionnaire assessment: Ll. 279-282 “one limitation…via video, for example)” discuss the limitation of using questionnaires. I would be happy to change this sentence to “One limitation of this study is that caregiver assessment of sleep was recorded, including assessment whilst caregivers themselves were asleep. This caregiver-questionnaire generated data could lead to a higher rate of inaccurate responses when compared to either direct interviewing or independent observation (via video, for example).” In addition, I could change Ll 308-309 “video or audio recording of dogs sleeping in their home environment may be optimal” to “Direct observation via video or audio recording of dogs sleeping in their home environment may be optimal”.

Regarding more detailed and accurate assessment of the problem behaviours, I have suggested the following change in response to reviewer 1’s feedback, which I hope will also resolve this point.

I suggest a change to Ll. 295-298 “PBS is not a validated…behavioural medicine consultations” to: “PBS is not a validated assessment tool used for research, thereby limiting direct comparison with previous work. It is based on a single item questionnaire answer. Free text responses regarding the nature of the problem behaviours eliciting the PBS were encouraged in the caregiver questionnaire and in the authors’ opinion the majority of these responses correlated with the reported PBS. Qualitative analysis of these free text answers was not possible due to an insufficient level of detail. Future research into this area would benefit from a more detailed assessment of presenting problem behaviours. PBS remains a valid choice in this initial exploratory study as a clinically relevant factor to both the individual patient and their caregivers.”

With regards to your concern on the potential discrepancy between “a real behavioural problem” and “a behaviour considered as problematic by owners”, I request that the paragraph Ll 372-386 is considered. If further qualification is required, please let me know if there are specific points you wish me to raise in addition to those discussed in this section.

Re: the point you raised on the study population and types of problem behaviour:

The caregivers answering the questionnaire in this study were recruited from the general public. They include both caregivers who perceive their dogs show problem behaviours and caregivers who believe their dogs show no problem behaviours (as evidenced by the wide range in PBS).

As this was an initial exploratory study, it was not possible to assess the type of problem behaviour, or the motivations for these behaviours. However, I agree that this would be beneficial in the future and I hope the suggested insertion of the paragraph above (Ll. 295-298 “PBS is not a validated assessment tool used….”) is sufficient to address this concern.

Round 2

Reviewer 3 Report

The authors addressed the main issues reported in the previous review. However, I would suggest indicating at line 492 "Future research into this area would benefit from a more detailed assessment of presenting problem behaviours..." this sentence:
"..., considering that owner evaluation of their dog behaviour may not reflect the presence of a real behavioral problem but rather a behaviour considered as problematic (d’Ingeo et al. 2022: Motivations of Human Helping Behavior towards Dogs. Veterinary Sciences, 9(3),145, and d’Angelo et al. 2020: Human-animal relationship dysfunction: A case study of animal hoarding in Italy. Animals, 10(9), 1501)."

Author Response

Thank you for the recommendation. For clarity; in the version of the article I have downloaded from the MDPI page, the sentence you refer to “Future research…” begins in Ll 303.

I have read the two cited papers in detail.

D'Ingeo et al. (2022) investigates and discusses the human motivations for dysfunctional helping behaviours such as “capture of non-socialized dogs or superficial/not suitable adoptions”. Whilst  “behavioural problems (ie. anxiety, phobia and aggression)” are briefly mentioned, this is not a caregiver-reported measure and the discussion of caregiver perception vs the diagnosis of a behavioural medicine clinician is not discussed. Whilst I feel this paper contributes valuable information to the field of veterinary behaviour, I do not feel it discusses the point you are requesting I include and as such I feel a citation is not appropriate.

D’Angelo et al. (2020) describes a severe hoarding case and the human motivations (of the individual “Mrs P”) for this behaviour along with an analysis of how this was managed by authorities. A very brief description of the hoarded animal’s emotional responses and behaviours (mentioned in three short comments in 3.1; “wary of human manipulation and fearful towards sudden noises or gestures” along with “discomfort and insecurity towards humans” and lastly “coprophagy”) is included, but this is not compared with Mrs P’s assessment of these responses. Figure 3 presents a list of behavioural problems but does not report which individuals recorded or classified these behaviours or whether the perception of Mrs P was recorded. In 3.2 and 4.1 the presenting behavioural problems are described further, with suggestions for the causes of these behaviours voiced by the authors. The perception of Mrs P, of the problem behaviours the animals show, is not reported. A comment within the discussion presents the concept of “dissociation” where the hoarder cannot perceive the reality of their own, and therefore their animals’, living conditions. This section does not discuss the hoarder’s perception of the behaviour of the animals. In my view this paper is hugely important and succeeds in its aim to “give emphasis to the emerging problem of animal hoarding” however it, at no point, compares caregiver perception of problem behaviours shown by the animals to the diagnosis of a behavioural medicine clinician and as such I feel citation is not appropriate.

Both of these cases illustrate significant cognitive dissonance between the benevolent intensions of the humans involved and the impact those humans were having on the quality of life of the animals they were involved with. In my opinion, neither case discuss that “owner evaluation of their dog behaviour may not reflect the presence of a real behavioral problem but rather a behaviour considered as problematic”. I again suggest that this point is covered comprehensively in the full paragraph Ll 380-394 (this was referred to as Ll 372-386 in the first response to reviewers) “It is important to consider whether caregiver perception (generating PBS) should be the measure by which we assess the potential link between sleep and emotional health…”.

I have further emphasised this point with a minor addition, as suggested in Ll 304-305 to read “Future research into this area would benefit from a more detailed assessment of presenting problem behaviours; to include clinician assessment in addition to caregiver perception of the presenting problems”.